# Standards-based audit to improve quality of maternal and newborn care—A stepped-wedge cluster randomised trial in Malawi

**Sarah Ann White**[ID]*, **Florence Mgawadere**[ID]¤a, **Somasundari Gopalakrishnan**¤b, **Nynke van den Broek**¤c

Liverpool School of Tropical Medicine, Liverpool, United Kingdom

¤a Current address: Institute of Population Health, School of Health Sciences, University of Liverpool, Liverpool, United Kingdom
¤b Current address: Independent Consultant, Maternal Epidemiology, Birmingham, United Kingdom
¤c Current address: Senior Health Advisor, Expertise France, Paris, France
* sarah.white@lstmed.ac.uk

## Abstract

### Background

Audit is a quality improvement approach used in maternal and newborn health. Our objective was to introduce the practice of standards-based audit at healthcare facility level, and to examine if this would improve quality of care assessed by compliance with standards developed and agreed with healthcare providers. Our focus was on emergency obstetric and newborn care (EmONC).

### Methods

A multidimensional incomplete stepped-wedge cluster randomised trial with 8 steps was conducted in 44 healthcare facilities in Malawi. A total of 25 standards of care were developed. At each healthcare facility one (health centres) or two (hospitals) standards were audited per cycle with two consecutive audit cycles conducted. Each cycle consisted of five steps: (i) select standard to be audited, (ii) measure compliance with standard (measurement 1), (iii) review findings and identify what changes are required to increase compliance (iv) implement changes, (v) re-measure compliance (measurement 2). Each compliance measurement assessed 25 women. Multilevel mixed effects logistic regression models were used to analyse data for all standards.

### Results

The crude overall compliance rate rose from 45% in the control phase (measurement 1) to 63% in the intervention phase (measurement 2) (from 51.6% to70.6% at Basic and from 34.5% to 50.8% at Comprehensive EmONC healthcare facilities. When adjusted for standard, facility type, month, and healthcare facility by month, the adjusted OR (95% CI) was 2.80 (1.65, 4.76). Actions taken to improve compliance with standards included improving staff performance of clinical duties and general conduct through re-orientation and staff

**Data Availability Statement:** Data are available from the LSTM online archive using https://doi.org/10.57978/lstm-q70s-qr85.

**Funding:** NvdB received funding for this trial from UNICEF Malawi under a grant entitled Improving the Quality of Care for Mothers and Babies (01/01/2014 to 15/02/2019) the funders had no role in study design, data collection and analysis, decision to publish, or preparation of the manuscript.

**Competing interests:** The authors have declared that no competing interests exist.

meetings as well as improved supervision, and, ensuring basic equipment and consumables were available on site (thermometers, rapid diagnostic tests, partograph).

## Conclusion

The introduction of standards-based audit helped healthcare providers identify problems with service provision, which when addressed, resulted in a measurable and significant improvement in quality of care.

## Trial registration

ISRCTN registration number: 59931298.

## Introduction

The latest estimates on maternal and child mortality show that globally, each year, an estimated 300,000 women die during or after pregnancy and childbirth, three million stillbirths and 2.4 million neonatal deaths occur [1,2]. Neonatal deaths account for 47% of all deaths in children under five years of age. Most of these deaths could have been prevented if actions that are proven to be effective had been in place.

Significant progress has been made with increased global coverage of interventions to improve maternal and newborn health and by 2020 up to 84% of births globally are assisted by skilled healthcare personnel, including medical doctors, nurses, and midwives which represents an increase of about 20 percentage points compared to 2001–2007 [3]. However there is wide recognition that further improvement in outcomes will depend on the ability to address the gap between coverage and quality [4,5].

Maternal, new-born and child health are inextricably linked–the survival and health of the new-born baby depends on the health of the mother during and after pregnancy and on the care she receives. Effective interventions at the time of childbirth and the period immediately after birth are particularly critical to reduce maternal deaths and morbidity, stillbirths and early neonatal deaths and morbidity–the 'triple return'. Skilled Birth Attendance, Emergency Obstetric Care, and Essential (or early) Newborn Care are internationally defined care bundles which reduce maternal, neonatal mortality and stillbirths.

A skilled birth attendant is 'an accredited health professional—such as a midwife, doctor or nurse—who has been educated and trained to proficiency in the skills needed to manage normal (uncomplicated) pregnancies, childbirth and the immediate postnatal period, and in the identification, management and referral of complications in women and new-born babies' [6]. In 1997, the collective minimum set of interventions required were bundled into a package of key interventions (or 'signal functions') known as Emergency Obstetric Care. In 2009 an additional intervention was added: to be able to perform basic neonatal resuscitation with a bag and mask. [7,8] Seven signal functions, namely, providing parenteral oxytocin, anti-convulsants (magnesium sulphate) and antibiotics, manual removal of a retained placenta, assisted vaginal delivery, removal of retained products of conception after miscarriage if required and resuscitation of the newborn, constitute Basic Emergency Obstetric and Newborn Care (BEmONC). These and two others–availability of caesarean section and ability to provide a blood transfusion–constitute Comprehensive EmONC. Internationally agreed criteria for minimum coverage levels needed at population level are that for a population of 500,000, there

should be at least one healthcare facility able to provide the nine components (signal functions) of Comprehensive EmONC (usually a hospital) and at least four other healthcare facilities that can provide the seven components (signal functions) of Basic EmONC (a hospital or health centre). (S1 Table) [7,8].

For care to be effective it must be evidence-based and of good quality. This requires that women and their families are treated with dignity and respect, care is people-centered, safe, effective, timely, efficient, and equitable [9–11]. Various audit approaches have long been used as part of clinical practice in maternal and newborn health, including maternal and perinatal death review and standards-based or clinical audit [12–14]. All approaches aim to assess the standard or quality of care and to identify what needs to be done to improve this. The National Institute for Clinical Excellence (NICE) defines standards-based audit as a quality improvement process that seeks to improve patient care and outcomes by systematically reviewing care against explicit standards, with identification and implementation of changes needed to achieve the desired standard of care [15]. A standard generally means a measure, norm, or model that can be used in comparative evaluations. In standards-based audit a standard consists of a measurable stated objective with: structure (what is needed), process (what needs to be done) and outcome (what will be the result achieved) criteria. Without agreed measurable standards of care, judgment on the quality of care can only remain subjective.

The World Health Organisation (WHO) working with international experts led the development of standards for maternal and neonatal health for adaption and adoption by countries [9]. Countries can adapt and expand these standards in line with national guidelines for service organisation and clinical practice and adopt these as part of national quality of care strategies and frameworks for the delivery of maternal and newborn health services.

Standards-based audit uses a participatory and systematic self-assessment approach. A complete audit cycle comprises of five steps: (i) agree a standard, (ii) measure practice i.e. compliance with standard (measurement 1), (iii) review findings and identify what changes need to be made to be able to meet the standard agreed (iv) implement changes, (v) re-measure practice i.e. compliance with standards (measurement 2).

A Cochrane review on the effectiveness of audit and feedback shows that this has potential to measurably improve quality of care including through increase in healthcare providers' compliance with desired practice. There is however an identified research 'gap' and need to establish if and how standards-based audit can be used to measurably improve quality of clinical practice by health care providers in maternal and newborn health [16]. We conducted a stepped-wedge cluster randomised trial to test the hypothesis that the process of standards-based audit improves quality of care by increasing the level of compliance with standards for maternal and newborn care. Our focus was on standards for Emergency Obstetric and Newborn Care.

## Methodology

### Study design

A four-dimensional incomplete cross-sectional stepped wedge cluster randomised trial [17] was conducted between 1st July 2018 and 7th March 2019. This design was selected because a clinical audit involves both a control (step (ii)) and intervention (step (v)) phase of data collection. Clinical audit is conducted using samples of patients within a healthcare facility; thus, to evaluate the impact of conducting standards-based audits, after introduction of the process at a healthcare facility, clustering by facility was necessary. The dimension was determined by the maximum number of standards to be audited within each healthcare facility. Data for each

standard assessed was collected for a cluster of individuals within a subset of steps at participating healthcare facilities (clusters), the design was therefore 'incomplete'.

## Trial intervention

The intervention consisted of the introduction of a process of standards-based audit consisting of 3-month audit cycles, each of which comprised: i) a month in which data were collected to assess compliance with the standard(s) chosen for audit (measurement 1); ii) an 'action month' in which the required identified actions were implemented to improve compliance with the standard; and, iii) a third month in which data were collected to re-assess compliance with the standard(s) (measurement 2). For each standard that was audited at a facility the 'action month' of the audit cycle was the step in which the classification of the status transitioned from control to intervention.

## Trial outcomes

The primary outcome measured for each woman was compliance with defined standard of care without regard to which standard was audited. Each secondary outcome measure was the same for a specified standard.

## Study setting

The Ministry of Health in Malawi identified five of its 28 districts for the implementation of a quality improvement programme (opportunistic selection). In each district all healthcare facilities providing Comprehensive or Basic EmONC were selected. A total of 44 facilities were included with 34 providing Basic (BEmONC, health centres, primary level) and 10 providing Comprehensive Emergency Obstetric Care (CEmONC, hospitals, secondary level) in the five implementation districts (S2 Table).

Each participating healthcare facility was to implement two consecutive periods of standards-based audit with each period anticipated to take three months. To avoid confounding in the design the steps in which audit cycle periods commenced were staggered over 3 months (steps); the design thus involved six steps. Randomisation was balanced for each standard, district, and facility type. Fifteen facilities were randomised to start in each of months 1 and 3 and the other 14 were to start in month 2.

CEmONC facilities conducted audit on two standards per quarter (four standards each in total) and BEmONC facilities audited one standard in each quarter (two standards each in total).

## Development of standards

A total of 25 standards of care, including standards for respectful care and for clinical content specific to EmONC, were developed and agreed (S3 Table). Based on existing WHO guidelines and the internationally agreed components of EmONC, these standards were developed through discussion and consensus agreement at five multidisciplinary consensus building national and international workshops (including managers, midwives, general doctors, obstetrician-gynaecologists, and researchers).

## Training in standards-based audit

Healthcare providers from each facility participated in a four-day participatory training workshop on how to conduct standards-based audit, how and which data to collect, including the design of simple audit data collection tables and before and after comparison of

percentage compliance. The healthcare providers from each facility worked together to identify relevant priority standards for their healthcare facility from the list of 25 standards provided, with the understanding that any standard chosen was expected to be applicable to at least 25 clients per month at the relevant healthcare facility. Participants from CEmONC facilities identified six and participants from BEmONC identified four standards in total.

### Assigning standards

The trial statistician (SW) subsequently assigned the standards to be audited by each facility to ensure coverage of as many standards of interest as possible, drawn from the standards ranked for each healthcare facility (where possible facilities that ranked a standard first were selected for the standard). The sequence in which standards were to be audited within each facility was assigned by the trial statistician using the **runiform**() function in a Stata program. For each standard the design of the study was an incomplete stepped wedge design. Thus, for each of the facilities assigned to audit a given standard, the timing of assessments (measurement 1 and measurement 2) for a standard was assigned to follow one of the six sequences listed for a typical standard 'a' (S1 Fig). For CEmONC facilities that audited four standards the design was four-dimensional. The dimensions do not correspond with standards. Allocation concealment was not possible for this trial's design.

### Data collection

Each standard was assessed during the first and third month of an audit cycle (S1 Fig). In addition, compliance with all standards selected for that facility was measured in the first and final (sixth) months of data collection at each healthcare facility. For example, S1 Fig indicates all assessments that were to be performed for standards audited within BEmONC facilities auditing standard 'a'. For each standard a standard-specific data collection tool was developed to capture the data required to determine whether the standard had been achieved in the care provided for each patient assessed. For some standards an exit interview was conducted to collect data (eg respectful care); for other standards data were collected from registers or case notes. At each healthcare facility data for a standard was collected by the trained healthcare providers at the facility for 25 consecutive women attending for care within the relevant month. No patient identification details were captured for any standard. For each audit cycle the healthcare providers kept a log of details of the problems identified and actions undertaken to improve compliance with the standard.

### Sample size considerations

Sample size calculations were performed using an R Shiny app to determine differences in compliance detectable for individual standards [18]. It was assumed that data for each standard would be collected according to the incomplete stepped wedge design shown in S1 Fig, with three clusters following each sequence in the design. For assessment of compliance with each standard, samples of 25 clients per standard, month and facility were planned. The calculations performed indicated that there would be at least 80% power to detect an absolute improvement in compliance of 22% (or an odds ratio of 2.6) for a standard audited at 18 healthcare facilities, provided the initial compliance was about 50%, the intra-cluster correlation did not exceed 0.2 and the cluster autocorrelation did not fall below 0.5.

## Statistical analysis

As specified in the statistical analysis plan data aggregated across standards were analysed, using Stata version 14, to provide an estimate of the impact of performing standards-based audit on compliance with standard. Analysis used all data collected for each standard and was on an as implemented basis. Analysis used the binary responses (compliant / not) of respondents within facilities in multilevel mixed effects logistic regression models, with fixed effects for Intervention (measurement 2 vs measurement 1), Standard (12 standards), Facility Type (Basic and Comprehensive) and Month (calendar month) (the statistical analysis plan incorrectly stated that Month would be a random effect), and random effects for Facility (cluster), and Month by Facility interaction (model extension B described by Hemming and colleagues) [19]. District was also considered for inclusion as a random effect. The effect of performing standards-based audit on compliance was estimated with a 95% confidence interval, after adjusting for Standard, Facility type, Month and Facility by Month interaction. Each standard considered was also to be analysed separately, to provide an estimate of the effectiveness of audit on compliance with the standard. These analyses involved the same terms as for aggregated data, except for terms involving Standard.

Several sensitivity analyses were performed to explore the impact of assumptions made when planning the trial analysis:

i). To examine whether there was any statistical evidence that the baseline compliance rate influences the odds of improvement following the action of the audit cycle the models used were expanded to include baseline compliance (on the percentage scale) as a covariate and interaction of this with the exposure variable. This analysis excluded the five facility-standard combinations for which no baseline data were collected.

ii). To examine evidence of any impact of overlap in the content of Standards an indicator factor was added to indicate data collected in period 2 for a standard which overlapped with a standard audited in period 1 at the facility.

iii). The planned analysis used only one two-way interaction term in the analysis. Five two-way interaction terms (Intervention by Facility Type, Standard by Facility Type, Standard by Month, Facility Type by Month and healthcare facility by Standard) were each considered for inclusion along with baseline compliance, in separate analyses to explore possible model extensions. These were applied to all eligible data and to the subset of data for the four most audited standards.

iv). To assess the impact on the estimates of the inclusion of additional assessments in month 6 (or 1) when audited in period 1 (or 2) two approaches were used: i) the mean (SD) of the change in compliance within the same intervention state (months 3 and 6 when audited in period 1 and months 1 and 4 for audits in period 2) was considered; ii) the main analysis was repeated using only data from audit cycles, i.e. with the additional assessments excluded.

## Ethical considerations

Ethical approval was obtained from the Research and Ethics Committee, Liverpool School of Tropical Medicine (Research Protocol 18–028, approved 21[st] June 2018). The Ministry of Health and Population, Malawi authorised conduct of the study waiving ethical approval in country (letter dated 20[th] June 2018, ref QMD/10). As consent is not normally part of a clinical audit process consent was not sought.

## Results

All 44 facilities completed the trial, but two facilities deviated from the intended schedule for audit cycles: participation in the training workshop was delayed. In both cases they were trained with sequence 3 and advised to complete only the second planned audit cycle, in order to adhere to the randomisation, but both chose to transfer to follow sequence 3 (Fig 1). Some facilities chose to audit a standard they had not been assigned to audit. Thus the partograph standard was audited by 24 facilities rather than the 18 planned. In some instances the standard selected was subsequently deemed not to be appropriate and an alternative standard was chosen. Reasons for no longer being appropriate included a lack of clients (e.g. women who have a APH or PPH for the Hb after APH / PPH standard); lack of reagents (e.g. urine testing involved for the identification of pre-eclampsia and ANC pre-eclampsia screening standards); good compliance (e.g. partograph for the partograph standard). The numbers of facilities assigned to each standard and the numbers actually assessed for each standard are summarised in S4 Table.

Details of the deviations from the assigned standards and randomised sequences are summarised in S5 Table. One hospital selected the ANC pre-eclampsia screening standard however since clients are required to pay for urine testing it was not appropriate to include data from this facility for this standard in the study.

The four most frequently selected standards were audited in 14 or more healthcare facilities (standards for respectful care, sepsis detection, malaria detection and monitoring of labour using a partograph). For three standards (Hb after APH / PPH, identification, and management of pre-eclampsia) there were only one or two health care facilities which completed data collection for an audit cycle. For the other five standards chosen and subjected to standards-based audit, there were between three and nine healthcare facilities which completed an audit cycle.

For eight of the 12 standards audited, the mean compliance rate was below 50% in the control state (Table 1). The exceptions were the oxytocic standard with a mean compliance (from 9 healthcare facilities) of 96%, malaria detection standard with a mean compliance (14 healthcare facilities) of 73%, Hb after APH / PPH standard which was only audited in one healthcare facility with 62% compliance at baseline, and ANC pre-eclampsia screening standard which was audited in three healthcare facilities with 67% compliance at baseline.

Overall, aggregating across all 12 standards used, 104 audit cycles were completed, 56 in the first period and 48 in the second one, with 3,140 client visits assessed in the control state and 3,276 in the intervention state. For some audit cycles fewer than the planned 25 were available; the mean (SD) number of clients per month was 20.8 (6.9). Across all health care facilities combined, the compliance rate (number of observations) increased from 45% (N = 3,140) to 63% (N = 3,276), with a crude OR (95%CI) of 2.09 (1.89,2.32) (Table 1). The compliance (number of observations) increased from 51.6% (N = 1,973) to 70.6% (N = 2,073) at healthcare facilities providing BEmONC and from 34.5% (N = 1,167) to 50.8% (N = 1,203) at healthcare facilities providing CEmONC.

### Primary outcome: Aggregated standards

For the primary outcome, using data aggregated for all standards, the improvement in compliance attributed to the use of standards-based audit was statistically significant; the adjusted (for standard, facility type, month, and healthcare facility by month clustering) OR (95% CI) was 2.80 (1.65,4.76) (Table 2). The ICC (95% CI) for healthcare facilities was estimated to be 0.14 (0.04,0.25) and for the interaction of healthcare facility and month the ICC was estimated to be 0.42 (0.34,0.49). Consequently, the CAC was estimated to be 0.34.

Facilities assessed eligibility (35 Bs and 9 Cs)[a]

Randomised (35 Bs and 9 Cs)

| Sequence 1 Facilities allocated (n=12 B + 3 Cs)[b] 1 C downgraded to B | Sequence 2 Facilities allocated (n=11 Bs +3 Cs) | Sequence 3 Facilities allocated (n=12 Bs +3 Cs) 2 Bs upgraded to C |
|---|---|---|
| **Month 0** | | |
| Completed training workshop (n=12 Bs + 2 Cs). 1 B completed training workshop with sequence 3 in month 2 and moved to sequence 3[c] | Completed training workshop early (n=1 B) | |
| **Month 1** | | |
| Assessed compliance to standards for both ACs (n=12 Bs + 2 Cs; 28 Stds; mean(sd)=22.1(5.7)) | Completed training workshop (n=9 Bs + 3 Cs) 1 B completed training workshop with sequence 3 in month 2 and moved to sequence 3[c] | |
| **Month 2** | | |
| Action taken to address deficiencies identified for first AC standard(s) (n=12 Bs + 2 Cs) | Assessed compliance to standards for both ACs (n=10 Bs + 3 Cs; 29 Stds; mean(sd)=20.6(7.3)) | Completed training workshop (n=12 Bs +5 Cs) 2 Bs moved into this sequence [c] |
| **Month 3** | | |
| Re-assessed compliance to first AC standard (n=12 Bs + 2 Cs; 16 Stds; mean(sd)=19.8(7.5)) | Action taken to address deficiencies identified for first AC standard(s) (n=10 Bs + 3 Cs) Re-assessed compliance to first AC standard (n=1 C;1 Std; 7 clients) | Assessed compliance to standards for both ACs (n=12 Bs + 5 Cs; 41 Stds; mean(sd)=21.7 (6.0)) |
| **Month 4** | | |
| Assessed compliance to second AC standard(s) (n=12 Bs + 2 Cs; 14 Stds; mean(sd)=21(6.7)) | Re-assessed compliance to first AC standard(s) (n=10 Bs + 3 Cs; 16 Stds; mean(sd)=19(7.9)) | Action taken to address deficiencies identified for first AC standard(s) (n=12 Bs + 5 Cs) 1 C also took action for standards assigned to second AC |
| **Month 5** | | |
| Action taken to address deficiencies identified for second AC standard(s) (n=12 Bs + 2 Cs) | Assessed compliance to second AC standard(s) (n=10 Bs +3 Cs; 14 Stds; mean(sd)=19.6(7.5)) | Re-assessed compliance to first AC standard(s) and 2 second AC standards acted on in month 4 (n=12 Bs + 5 Cs; 23 Stds; mean(sd)=20.6(7.4)) 1 C also assessed compliance for one second AC standard (1 Standard; 25 clients) |
| **Month 6** | | |
| Re-assessed compliance to both AC standards(s) (n=12 Bs + 2 Cs; 30 Stds mean(sd)=20.4(7.4)) | Action taken to address deficiencies identified for second standard(s) (n=10 Bs + 3 Cs) | Assessed compliance to second AC standard(s) (n=12 Bs + 4 Cs;18 Stds; mean(sd)=23.4(4.2)) 2 Cs also assessed compliance for one first AC standard (mean(sd)=25(0)) |
| **Month 7** | | |
| | Re-assessed compliance to standards for both ACs (n=10 Bs +3 Cs; 28 Stds; mean(sd)=20.7(7.6)) | Action taken to address deficiencies identified for second AC standard(s) (n=12 Bs + 4 Cs) |
| **Month 8** | | |
| | | Re-assessed compliance to both AC standards (n=12 Bs + 5 Cs; 42 Stds; mean(sd)=21.5(6.5)) |
| **Month 9** | | |
| | | Re-assessed compliance to standards (n=2 Bs; 2 Stds; mean(sd)=17(11.3)) |

**Fig 1. CONSORT Flowchart for stepped wedge cluster randomised trial by allocated sequence and period[ab].** a
B = BEmONC; C = CEmONC; AC = audit cycle. b n = numbers of BEmONC and CEmONC facilities assessed; s Stds = number of standards (s) for which data were collected; mean(SD) for numbers of clients assessed per standard. c one BEmONC facility was delayed in completing pre-trial induction; as this resulted in the facility being unable to complete the first audit cycle allocated they were advised to complete only the second audit cycle scheduled for their sequence; instead they proceeded to follow the schedule for Sequence 3. During training each facility ranked standards for importance to be audited; subsequently each facility was assigned the standard(s) to be audited in each audit cycle; each BEmONC was to do one standard per audit cycle while each CEmONC was to do two. S5 Table details the changes of standards used and the occasions on which data were intended to be collected but were not.

**Table 1. Summary statistics for each standard audited and for all standards combined, by study arm, with crude comparison.**

| Standard Audited | Stratum | No. of HCFs | | % Compliance (N) | | Crude difference | Crude OR |
|---|---|---|---|---|---|---|---|
| | | P1 | P2 | Control | Inter-vention | | |
| 1. All women attending for birth are received and treated with respect (respectful care) | I | 1 | 3 | 35.4 (633) | 54.3 (612) | 18.9 | 2.16 (1.71,2.74) |
| | II | 0 | 4 | | | | |
| | III | 5 | 4 | | | | |
| 2. As part of active management of the third stage of labour, all women giving birth at the healthcare facility receive an oxytocic (oxytocic) | I | 0 | 1 | 96.2 (262) | 97.4 (343) | 1.2 | 1.47 (0.53,4.16) |
| | III | 6 | 2 | | | | |
| 3. All women who have an antepartum (APH) or postpartum- haemorrhage (PPH) have their haemoglobin (Hb) checked and recorded (Hb after APH / PPH) | II | 0 | 1 | 61.5 (13) | 100.0 (5) | 38.5 | NA |
| 4. All women who have an uncomplicated birth at the healthcare facility have their temperature measured and recorded at least once after birth and before discharge (sepsis detection) | I | 4 | 3 | 30.2 (663) | 58.7 (695) | 28.5 | 3.29 (2.61,4.15) |
| | II | 4 | 2 | | | | |
| | III | 4 | 2 | | | | |
| 5. All women with fever are tested for malaria within 24 hours (malaria detection) | I | 1 | 3 | 72.7 (282) | 89.9 (276) | 17.2 | 3.33 (2.04,5.53) |
| | II | 2 | 3 | | | | |
| | III | 3 | 2 | | | | |
| 6. Every woman in labour has her blood pressure measured, urine tested for protein and the results recorded (identification of pre-eclampsia) | I | 1 | 0 | 35.7 (14) | 38.9 (18) | 3.2 | 1.15 (0.22,6.30) |
| | III | 1 | 0 | | | | |
| 7. Every woman with pre-eclampsia or eclampsia has a fluid input-output chart completed (management of pre-eclampsia) | II | 1 | 0 | 0.0 (2) | 100.0 (4) | 100 | NA |
| 8. All women attending for antenatal care have their blood pressure checked and urine tested for protein (ANC pre-eclampsia screening) | I | 0 | 0 | 66.7 (66) | 91.8 (49) | 25.1 | 5.63 (1.69,23.9) |
| | II | 2 | 0 | | | | |
| | III | 0 | 1 | | | | |
| 9. Every woman in labour is monitored using a partograph correctly (partograph) | I | 7 | 2 | 38.6 (770) | 54.2 (954) | 15.6 | 1.88 (1.55,2.30) |
| | II | 6 | 2 | | | | |
| | III | 2 | 5 | | | | |
| 10. All women who need an emergency Caesarean Section should be delivered within 60 minutes of the decision (CS timing) | I | 1 | 0 | 46.3 (108) | 38.7 (111) | -7.6 | 0.73 (0.41,1.30) |
| | III | 2 | 1 | | | | |
| 11. Every woman with an incomplete miscarriage/abortion undergoes evacuation / manual vacuum aspiration (MVA) within 24 hours of arrival at the healthcare facility (MVA timing) | I | 0 | 1 | 39.5 (76) | 79.7 (64) | 40.2 | 6.02 (2.65,14.0) |
| | II | 0 | 1 | | | | |
| | III | 1 | 0 | | | | |
| 12. Every woman who has had uterine evacuation or manual vacuum aspiration (MVA) has a clinical examination before being discharged home (MVA exam) | I | 1 | 1 | 42.2 (251) | 56.6 (145) | 14.4 | 1.78 (1.15,2.75) |
| | II | 1 | 2 | | | | |
| | II | 0 | 2 | | | | |
| All standards | | 56 | 48 | 45.3 (3,140) | 63.4 (3,276) | 18.1 | 2.09 (1.89,2.32) |

HCF = healthcare facility; P1 = period 1; P2 = period 2; N = number observed; OR = odds ratio.

A sensitivity analysis, excluding the three standards used in no more than two healthcare facilities, yielded similar results, with an adjusted OR (95% CI) of 2.92 (1.72,4.96). Further analyses considered omission of the random healthcare-facility by month effect and confirmed that the fit of the model was significantly improved when the term was included. Conversely inclusion of a random term for district did not significantly improve the fit of the model.

## Secondary outcomes: Compliance with individual standards

Analysis of data for the seven standards with sufficient data for a separate analysis yielded ORs which range from 1.10 (0.25,4.90) (for CS timing standard) to 19.6 (3.10,124) (for respectful

**Table 2. Estimates of compliance OR between study arms and correlations.**

| Standard | Secular trend included? | N | Adjusted[a] OR (95% CI) | ICC (HCF) (95% CI) | ICC (HCF * month) (95% CI) | CAC |
|---|---|---|---|---|---|---|
| All | Yes | 6,416 | 2.80 (1.65,4.76) | 0.14 (0.04,0.25) | 0.42 (0.34,0.49) | 0.34 |
| | No | 6,416 | 2.58 (2.28,2.91) | 0.20 (0.14,0.28) | NA | NA |
| Subset (excluding standards for Hb after APH / PPH, identification and management of pre-eclampsia) | Yes | 6,360 | 2.92 (1.72,4.96) | 0.14 (0.08,0.26) | 0.42 (0.34,0.49) | 0.35 |
| | no | 6,360 | 2.59 (2.29,2.92) | 0.20 (0.14,0.28) | NA | NA |

[a] adjusted for standard, facility type and month, with random effects for facility and facility-by-month interaction HCF = healthcare facility.

care standard) (S6 Table). The 95% confidence intervals for each of these covered the value of 2.80 estimated in the aggregate analysis. However, for four standards (ANC pre-eclampsia screening, partograph, CS timing and MVA exam) the 95% confidence interval included 1.0 and thus these analyses did not provide evidence that the observed increase in compliance was statistically significant. As the partograph standard was the standard most frequently audited, data for it were explored further.

The logistic regression model for the partograph standard estimated that a substantial secular increase in compliance occurred, with the OR for compliance compared with month one increasing to month eight. The distribution of month in which healthcare facilities took action for this standard was not well balanced, with just two facilities per step for steps 4, 5 and 6 but at least five facilities in each of the other three steps (2, 3 and 7). Compliance improved at all four healthcare facilities which took action in steps 4 and 5, between the baseline assessment and the initial assessment of the audit cycle (control state), these improvements would have contributed to the estimated upward secular trend. Additionally, compliance did not improve for the assessment after action in either of the healthcare facilities in step 5 (S7 Table and S2 Fig).

## Sensitivity analyses to explore robustness of model fitted

**Is the odds of improvement independent of the baseline compliance?.** Inclusion of the interaction between Baseline and Intervention after addition of the Baseline covariate gave a statistically significant improvement in fit ($X^2_1 = 37.2$, p<0.001), with a greater improvement in odds associated with lower baseline compliance.

**Does overlapping content of standard audited in period 1 impact compliance with standard audited in period 2?.** Three standards (identification of pre-eclampsia, ANC pre-eclampsia screening and partograph) involved measurement of blood pressure and urine, two standards (sepsis detection and MVA exam) measured temperature and two (Hb for APH / PPH and partograph) measured haemoglobin. There were seven instances at five healthcare facilities of one of these standards being audited in the second period following audit of an associated standard in the first period. A boost in the OR (95% CI) of compliance due to carry-over was estimated (1.94 (1.10,3.43)); the OR (95% CI) for the intervention increased slightly, with an increase in standard error.

**Additional interaction terms.** Analysis using the four standards most frequently audited (respectful care, sepsis detection, malaria detection and partograph) found statistically significant evidence of interactions of standards with each of, facility type (p = 0.02), Month

(p<0.001) and healthcare facility (p<0.001). No evidence was found of a significant interaction of facility type with either intervention (p = 0.29) or month (p = 0.65). In each analysis the estimated OR for the effect of the intervention remained very similar, or slightly increased in magnitude. When both the interaction of standard with month and standard with facility type were included the improvement in fit was statistically significant for standard by month but not for standard by facility type.

**Are additional baseline and endline assessments useful?.** The mean (SD) change in compliance between control months for 37 audit cycles in period 2 was 11% (30%). For the 30 audit cycles in period 1 with data at the end of period 2 the mean (SD) change between intervention months was -4% (36%). Although the levels had been expected to stay similar there was quite marked variability. Analysis with data for the additional assessments excluded yielded an estimated OR of compliance of (95% CI) of 3.29 (1.66,6.49).

## Problems identified and actions taken at healthcare facility as part of standards-based audit

A summary of the problems identified, and actions taken (steps 2 and 3 of the audit cycle) is presented in Table 3. The problems identified were classified into four categories:

A—Conduct and practice of healthcare providers—not carrying out duties as required, not aware of need for certain clinical practices and poor documentation of practice. This was noted as a problem by 29 facilities for 55 audit cycles, for all standards audited except the oxytocic standard. Staff shortage and/or non-availability of staff on site was a problem for 3 of 7 audit cycles for CS timing and MVA timing standards.

B—Lack of availability of equipment, consumables and tools for documentation was a problem reported at 18 (41%) facilities for 20 of the 69 audit cycles conducted for five standards (oxytocics, sepsis detection, malaria detection, ANC pre-eclampsia screening and partograph). This mainly required additional simple equipment and tools to be made available sometimes from existing stock (thermometers, urine dipsticks, partographs, patient charts).

C–General Infrastructure of the healthcare facility was a problem for 6 of 21 facilities auditing two standards (respectful care and CS timing). Examples were: the construction of a labour room not allowing for patient privacy e.g., open room with no separation between beds; no accommodation on site for emergency staff.

D–Patient- related factors were a recognised problem for 2 of 3 facilities meeting two standards (Hb after APH / PPH and identification of pre-eclampsia) and in each case simple diagnostic tests (measuring haemoglobin, checking protein in urine using a dipstick) could not be provided to women unable to pay i.e. these services were not free.

At healthcare facility level teams worked hard to solve these problems with practical and feasible solutions arrived at and implemented as illustrated for each standard in Table 3.

## Discussion

### Main findings

This study shows that for a pre-agreed set of standards reflecting the quality of care in maternal and newborn health, there was a statistically significant improvement in compliance for a standard audited. For a standard selected for audit the estimated OR (95% CI) of compliance with

**Table 3. Summary of problems identified, and actions taken to improve compliance with standards.**

| Standard (n/N) | Problems identified (number of healthcare facilities)–problem category[a] | Actions taken to resolve (number of healthcare facilities) |
|---|---|---|
| 1. All women attending for birth are received and treated with respect (9/17) | Healthcare providers do not wear uniform and cannot be recognised when on duty (3)—A<br>Healthcare providers do not greet clients or introduce themselves (3)—A<br>Labour rooms/area not designed to provide privacy (4)–C | Reorientation meeting with staff / staff to discuss importance of, and encourage uniform wearing (9)<br>Hospital in-charge to do spot checks (1)<br>Advocate for new labour ward (1) |
| 2. As part of active management of the third stage of labour, all women giving birth at the healthcare facility receive an oxytocic (6/9) | No space to write down the time for administering oxytocin on the partograph format used (6) -B | Orient staff where / how to document time on the partograph (6)<br>Ordered new partograph which allows for documentation of giving of oxytocin (1) |
| 3. All women who have an antepartum (APH) or postpartum- haemorrhage (PPH) have their haemoglobin (Hb) checked and recorded (1/1) | Hb sometimes only checked later i.e. >24 hours following haemorrhage (1)—A<br>Some women cannot afford to pay for blood tests (1)–D | Staff orientation to check all women who can pay before discharge (1) |
| 4. All women who have an uncomplicated birth at the healthcare facility have their temperature measured and recorded at least once after birth and before discharge (12/19) | Thermometers kept at home by healthcare providers, and they forgot to bring them to work (2)—B<br>No thermometers available (5)-B<br>Healthcare providers did not check temperature (11)—A<br>No charts to document observations available (3)–B | Staff re-orientation and reminders during handover meetings (11)<br>More thermometers ordered (4)<br>Standard chart introduced (1)<br>Thermometers provided (from stock) to the labour rooms and postnatal ward (2)<br>The facility in-charge intensified supervision (2)<br>Fortnightly spot checks by ward in charge (1) |
| 5. All women with fever are tested for malaria within 24 hours (9/14) | Practice not in place to conduct malaria testing for women with fever (9)—A<br>Malaria test kits out of stock at times (3)–B | Staff orientation (6)<br>Staff requested to measure whenever kits available (1)<br>Ordered more kits from DHO (6)<br>Kits made available at point of antenatal care service provision (3) |
| 6. Every woman in labour has her blood pressure measured, urine tested for protein and the results recorded (2/2) | Not routine practice to test urine for protein in all women (1)—A<br>Protein check not done routinely as women have to pay for this (1)–D | Meeting with all staff (1)<br>Importance of practice explained in meeting with all staff (2) |
| 7. Every woman with pre-eclampsia or eclampsia has a fluid input-output chart completed (1/1) | No chart attached to the woman's file or not completed (1)–A | Staff orientation (1) |
| 8. All women attending for Antenatal Care have their blood pressure checked and urine tested for protein (3/3) | Not routine practice to test urine for protein in all women (1)—A<br>Some women must be referred to another area of the healthcare facility as test kits available there only and not at point of providing antenatal care (3)—B | Staff meeting and oriented staff (2)<br>Urine test kits placed in the antenatal clinic (2) |
| 9. Every woman in labour is monitored using a partograph correctly (15/24) | Partograph supposed to be used to monitor labour, but healthcare providers are not doing this (14)—A<br>Partographs are used but incompletely filled (9)—A<br>Some staff deliberately don't use partographs (3)—A<br>Descent / labour progress is not documented on the partograph (4) -A<br>Action not taken on time when there is a complication (3)—A<br>Temperature was not checked or documented (2)—A<br>Respiration rate was not documented (1)—A<br>Partographs are not kept with the patient in labour and goes missing (1)—A<br>The colour of the liquor is not documented (1)–A<br>There are no thermometers available in the labour ward (1)–B | Meeting organised with the healthcare providers to re-orient everyone on why and how to complete a partograph (14) and including the need to take action following identification of a complication (4)<br>New thermometers ordered (1)<br>Thermometers made available from existing stock (1)<br>A designated area for keeping partographs identified (1)<br>'Spot checks' introduced and implemented by the person in charge of the labour ward (weekly/ fortnightly/ monthly) (7)<br>Monthly checking of partographs by the ward clerk (2) |

*(Continued)*

**Table 3.** (Continued)

| Standard (n/N) | Problems identified (number of healthcare facilities)–problem category[a] | Actions taken to resolve (number of healthcare facilities) |
|---|---|---|
| 10. All women who need an emergency Caesarean Section (CS) should be delivered within 60 minutes of the decision (3/4) | Failure of clinicians to record time decision for CS is made (3)—A<br> Difficult to find the clinician who is on call at night causing delays (1)–A<br> Clinicians not available or live far away (2)–A/C | Discussion with clinicians to stress importance of documenting time in patient records (3)<br> Planning to identify a designated room for on- call clinician to stay at the healthcare facility at night (1)<br> Clinicians requested to spend night in designated hospital room when on call (1) |
| 11. Every woman with an incomplete miscarriage/ abortion undergoes evacuation / manual vacuum aspiration (MVA) within 24 hours of arrival at the healthcare facility (2/3) | Shortage of staff sometimes (1)—A<br> Evacuation done late—after 24hrs (2)–A/C | Orientation with midwives & clinicians (1)<br> Agreed on improved prompt communication (1)<br> Requested more staff from DHO (1)<br> Designated one clinician to do the evacuation (1) |
| 12. Every woman who has had uterine evacuation or manual vacuum aspiration (MVA) has a clinical examination before being discharged home (6/7) | Not done by staff (2)–A<br> No guidance that it is important to document the examination in the client's notes (3)—A<br> Whether or not family planning advice was given is not documented (5)—A<br> Vital signs not always checked (3)—A<br> Patient did not receive a check for bleeding per vagina (2)—A<br> Family planning advice not provided (3)—A | Meeting to orient staff on the importance of doing a pre-discharge assessment (5) and to provide advice on family planning before discharge (2) and document this (1)<br> Meetings with staff to agree to conduct a complete assessment before discharge (1)<br> Developed a checklist for documentation and attached this to each client's notes (2) |

n = Number of healthcare facilities from which problems and actions were obtained; N = number that audited the standard.

[a] problem categories: A = Conduct and practice of healthcare providers; B = Lack of availability of equipment, consumables and tools for documentation; C = General Infrastructure of the healthcare facility; D = Patient- related factors.

standard, following feedback and action, in the intervention phase compared with the control phase was 2.80 (1.65,4.76).

For individual standards estimates of change in compliance (using ORs) were variable and prone to bias, likely due to the inherent confounding of time and intervention, and imbalance in the data available, which arose from deviations from the randomisation.

As part of the audit cycle healthcare providers identified problems that resulted in lack of compliance with the agreed standards of care. These were most frequently recognised as requiring a change in staff conduct and performance–with reorientation and improved supervision of staff implemented to improve clinical practice and documentation. Lack of consumables and equipment, e.g. thermometers, simple point-of-care diagnostic tests, partographs and patient charts, for five of the standards audited, was noted at 41% of facilities. In several healthcare facilities the overall infrastructure of the labour rooms required more attention as patient privacy could not be provided because of open labour rooms with no structural (or screens) division between beds. Non availability of staff for operative procedures (manual vacuum aspiration and caesarean section) was in part due to the non-availability of accommodation on site for staff required for emergencies after daytime working hours.

## Strengths and limitations

To the best of our knowledge this is the first study to assess the effectiveness of standards-based audit as a quality improvement method for Emergency Obstetric and Newborn Care using a stepped-wedge randomised trial design. The estimated impact of the intervention on the odds of compliance was consistent across the sensitivity analyses performed, suggesting that the findings for aggregated data are robust.

For some Standards the deviations from randomisation of both the allocated standards and their timings, created considerable imbalance in data obtained per step for the standard. The small number of observations for each step reduced the precision in estimation of secular trend. This reduced the power of the analysis for some individual standards to detect a benefit from the intervention and produced some imprecise estimates.

One standard (malaria detection) was seasonal since the incidence of malaria is usually higher during the later months of the study. It was audited at 15 healthcare facilities–six in period 1, nine in period 2. During the malaria season there may be a higher underlying chance of malaria being considered as the possible cause for fever and hence an increased chance of being tested for malaria as the study period progressed. If this is the case this could have caused the impact of the intervention to be over-estimated for some of the 15/104 audit cycles completed for this standard.

There was good overall balance in the design of data capture for the aggregated data analysed. Although an underlying upward trend in compliance was estimated, a statistically significant improvement in compliance was detected. In each study month at least 14 facilities contributed data to the aggregate analysis.

Inclusion of the baseline compliance rate as a covariate in the analysis improved the fit of models and is a useful strategy for accounting for part of the variation in compliance with the various standards. Although each healthcare facility contributed data for multiple standards there is little justification for assuming that compliance with all standards will be similar within any given healthcare facility. Indeed, it is anticipated that compliance for standards varies between healthcare facilities which were encouraged to choose standards for audit for which they perceived compliance to be sub-standard (rather than standards for which compliance was considered as already high).

## Implications for future trial design

The findings of the sensitivity analyses suggest some additional issues for consideration in planning the design and analysis for any future study using this class of trial design. These issues are itemised below:

1. Analysis including Baseline as a covariate suggests that the magnitude of the impact on the OR is greater when the initial compliance level is lower and therefore consideration should be given to inclusion of Baseline as a covariate.

2. The overlapping content of standards audited had minimal impact on the estimated impact of the intervention. However, because of the potential bias arising from such overlaps it would be prudent to avoid such overlaps in the selection of standards for auditing.

3. As is evident in the estimates derived from analysis of individual standards the sparseness of the design limits the capacity of the analysis to derive valid estimates for multiple effects. Although several interactions were statistically significant when included in an analysis the validity of such estimates is difficult to establish. Inclusion of interaction terms for standard by Month and standard by Facility Type separately indicated that each was statistically significant, but when included together the standard by Month interaction did not improve on the fit without the term in the model. This suggests that the inherent imbalance in the study design, with the distribution of months in which data are measured for each standard being heterogeneous, may be the cause of the apparent interaction when included without standard by Month rather than a real interaction between standards and Facility Type. Alternatively there may be a spurious association between the two interaction terms.

4. Exclusion of the additional baseline and endline assessments resulted in a slightly larger estimated OR for the impact of the intervention, with a decrease in precision. At some healthcare facilities the inclusion of the additional assessments was associated with an improvement in compliance between assessments without the intervention. The additional measurements provide opportunity to improve the precision and hence power of the study to detect an effect of the intervention.

## Implications for clinical practice

Improving the quality of healthcare services and making quality an integral component of scaling-up of interventions that are known to be effective is crucial if health outcomes for mothers and babies are to improve [4,5].

Skilled birth attendants need to be trained to have the required competencies and should be provided with an "enabling environment" which includes drugs, supplies, appropriate policies, and a functional referral system [20]. All women require skilled attendance at the time of birth [6,7]. For an estimated 10–15% of women, a potentially life-threatening complication develops during pregnancy, birth, or the post-partum period. In most cases, this complication will be unexpected and unpredictable. Therefore, it is crucial that all women and babies have access to good-quality Emergency Obstetric and Newborn Care [7,8]. The most frequent and main complications are well understood and can be readily prevented and/or treated. They include haemorrhage, sepsis, eclampsia, and complications of obstructed labour and abortion. There are existing effective medical and surgical interventions that are relatively inexpensive which can be put in place to manage these complications [21]. It is important that healthcare providers are confident and competent in the provision of this care [22–24]. Training of health care providers can improve implementation of evidence-based practice but for optimal effect the organisation and quality of health services require additional focus [25].

Multi-country surveys demonstrate that quality improvement activities are in place in many settings but that these tend to focus mainly on audit of adverse events through conduct of maternal and perinatal death audit [25,26]. This is generally effective especially in identifying specific areas of care that require improvement because they are assessed as sub-standard. Data on cause of and factors contributing to maternal and perinatal death are 'local' and can help identify specific gaps as well as solutions that can be put in place at the healthcare facility level [27,28]. This requires that a culture of improvement and quality is developed rather than a culture of blame. Specific standards can be agreed that reflect the identified areas of care that need to be improved. Systematic use of standards-based audit with healthcare providers themselves evaluating the care they are giving and making improvements where needed can lead to demonstrable improved motivation, ownership, and sense of responsibility for delivering good quality care [29,30].

Poor quality care is often a function of weak health systems and processes or problems in their implementation generally rather than the fault of individuals. Systematic reviews suggest that effectiveness is likely to be higher when baseline compliance is low, when feedback or audit cycles are conducted more than once, with local expertise in conduct of audit, team effort rather than individual actions with the agreement of explicit actions and development of action plans [16,29,30]. Support is also needed at local and central government levels to enable further scale-up audit.

## Supporting information

**S1 Fig. Schematic representation of one dimension of the stepped wedge, cluster Randomised trial implementation for standard 'a' showing measurements of compliance for all**

**standards audited at BEmONC facilities auditing standard 'a', by sequence and month of study.** 0() denotes assessment of compliance with the Standard(s) listed under the current standard of care. A() denotes action taken for standard listed, after review of data collected previously. 1() denotes assessment of compliance after taking action to improve the quality-of-care delivery for the standard(s) listed. Standards are indicated by letters *a*, *b*. . . *g*. Standard a is distinct from standards *b.. g*. Standards *b.. g* may all be distinct but were not required to be distinct. When a sequence was used for multiple facilities, the standard audited in the alternative period was not required to be consistent, though a single letter is used to denote it within this figure. Within each sequence the audit cycle periods are indicated by boldly bordered sets of three-month periods.
(TIF)

**S2 Fig. Spaghetti plots showing monthly percent compliance for partograph Standard, by facility type and period in which audited.** P1 = period 1; P2 = period 2.
(TIF)

**S1 Table. Essential components (signal functions) of Comprehensive and Basic Emergency Obstetric and Newborn Care (CEmONC and BEmONC).**
(DOCX)

**S2 Table. Participating healthcare facilities by district and level of service provision showing strata to which randomised.**
(DOCX)

**S3 Table. Standards of Care for Emergency Obstetric and Newborn Care (EmONC) developed through consensus by multidisciplinary group of healthcare providers, managers, and researchers[a].**
(DOCX)

**S4 Table. Numbers of HCFs to which each standard was assigned for audit, and at which audit was completed, by stratum and by period.**
(DOCX)

**S5 Table. Summary of deviations, in sequence and period when Standards were audited, from randomised standards, by facility.**
(DOCX)

**S6 Table. Estimates of compliance OR between study arms and correlations for each standard.**
(DOCX)

**S7 Table. Underlying secular trends for partograph and sepsis detection standards and the aggregate analysis.**
(DOCX)

**S1 File. Using standards-based audit to improve maternal and new-born health in Malawi —Study Protocol.**
(DOCX)

**S2 File. CONSORT checklist.**
(PDF)

## Acknowledgments

We would like to thank the healthcare providers at participating healthcare facilities who worked so hard to implement the audit and contributed to the study. We gratefully acknowledge the support of our colleagues of the Ministry of Health and Population in Malawi.

## Author Contributions

**Conceptualization:** Sarah Ann White, Nynke van den Broek.

**Data curation:** Somasundari Gopalakrishnan.

**Formal analysis:** Sarah Ann White.

**Investigation:** Florence Mgawadere, Nynke van den Broek.

**Methodology:** Nynke van den Broek.

**Supervision:** Florence Mgawadere, Somasundari Gopalakrishnan, Nynke van den Broek.

**Writing – original draft:** Nynke van den Broek.

**Writing – review & editing:** Sarah Ann White, Florence Mgawadere, Somasundari Gopalakrishnan.

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
