## [Decision Letter · Decision Letter 0]

2 Jul 2024

PONE-D-23-41107Standards-based audit to improve quality of maternal and newborn care – a stepped-wedge cluster randomised trial in MalawiPLOS ONE

Dear Dr. White,

Thank you for submitting your manuscript to PLOS ONE. After careful consideration, we feel that it has merit but does not fully meet PLOS ONE’s publication criteria as it currently stands. Therefore, we invite you to submit a revised version of the manuscript that addresses the points raised during the review process.

Please submit your revised manuscript by Aug 16 2024 11:59PM.  If you will need more time than this to complete your revisions, please reply to this message or contact the journal office at plosone@plos.org. Please include the following items when submitting your revised manuscript:A rebuttal letter that responds to each point raised by the academic editor and reviewer(s). You should upload this letter as a separate file labeled 'Response to Reviewers'.A marked-up copy of your manuscript that highlights changes made to the original version. You should upload this as a separate file labeled 'Revised Manuscript with Track Changes'.An unmarked version of your revised paper without tracked changes. You should upload this as a separate file labeled 'Manuscript'.If applicable, we recommend that you deposit your laboratory protocols in protocols.io to enhance the reproducibility of your results. Protocols.io assigns your protocol its own identifier (DOI) so that it can be cited independently in the future. For instructions see: https://journals.plos.org/plosone/s/submission-guidelines#loc-laboratory-protocols. Additionally, PLOS ONE offers an option for publishing peer-reviewed Lab Protocol articles, which describe protocols hosted on protocols.io. Read more information on sharing protocols at https://plos.org/protocols?utm_medium=editorial-email&utm_source=authorletters&utm_campaign=protocols.

We look forward to receiving your revised manuscript.

Kind regards,

Mergan Naidoo, PhD

Academic Editor

PLOS ONE

Journal Requirements:

2. We note that you have selected “Clinical Trial” as your article type. PLOS ONE requires that all clinical trials are registered in an appropriate registry (the WHO list of approved registries is at https://www.who.int/clinical-trials-registry-platform/network/primary-registries" https://www.who.int/clinical-trials-registry-platform/network/primary-registries and more information on trial registration is at http://www.icmje.org/about-icmje/faqs/clinical-trials-registration/). Please state the name of the registry and the registration number (e.g. ISRCTN or ClinicalTrials.gov) in the submission data and on the title page of your manuscript. a) Please provide the complete date range for participant recruitment and follow-up in the methods section of your manuscript. b) If you have not yet registered your trial in an appropriate registry, we now require you to do so and will need confirmation of the trial registry number before we can pass your paper to the next stage of review. Please include in the Methods section of your paper your reasons for not registering this study before enrolment of participants started. Please confirm that all related trials are registered by stating: “The authors confirm that all ongoing and related trials for this drug/intervention are registered”. Please see http://journals.plos.org/plosone/s/submission-guidelines#loc-clinical-trials for our policies on clinical trials.

"NvdB received funding for this trial from UNICEF Malawi under a grant entitled Improving the Quality of Care for Mothers and Babies (01/01/2014 to 15/02/2019) "

Reviewers' comments:

Reviewer's Responses to Questions

**Comments to the Author**

1. Is the manuscript technically sound, and do the data support the conclusions?

Reviewer #1: Partly

Reviewer #2: Yes

Reviewer #3: Partly

2. Has the statistical analysis been performed appropriately and rigorously? 

Reviewer #1: Yes

Reviewer #2: I Don't Know

Reviewer #3: I Don't Know

3. Have the authors made all data underlying the findings in their manuscript fully available?

Reviewer #1: Yes

Reviewer #2: Yes

Reviewer #3: Yes

4. Is the manuscript presented in an intelligible fashion and written in standard English?

Reviewer #1: Yes

Reviewer #2: Yes

Reviewer #3: No

5. Review Comments to the Author

Reviewer #1: A stepped-wedge cluster randomized trial was conducted at 44 healthcare facilities which aimed to assess clinical standards-based audits in improving maternal and newborn health as measured by compliance rates. Audited compliance standards improved significantly.

Minor revisions:

1- Line 312: Provide a measure of dispersion for the mean of 20.6.

2- Table 1: Define all abbreviations.

3- Indicate if an effect for the repeated measures from individual patients was included in the logistic regression. These correlated measures need to be accounted for in the models.

4- Line 378: The common abbreviation for standard deviation is SD.

5- Cite the statistical software used for the analysis.

Reviewer #2: Standards-based audit to improve quality of maternal and newborn care – a stepped- wedge cluster randomised trial in Malawi

Summary of the Research

Audit is a quality improvement approach used in maternal and newborn health. The objective of the study was to introduce the practice of standards-based audit at healthcare facility level, and to examine if this would improve compliance with standards of care developed and agreed with healthcare providers. The focus was on emergency obstetric and newborn care in a low resource setting. A multidimensional incomplete stepped-wedge cluster randomised trial with 8 steps (months) was conducted in 44 healthcare facilities in Malawi. A total of 25 standards were developed. Each facility had two consecutive audit cycle periods in each of which one (health centres) or two standards (hospitals) were audited in each period. Each audit cycle consisted of five steps: (i) agree the standard to be audited, (ii) measure compliance with standard, (iii) review findings and identify what changes are required to increase compliance (iv) implement changes, (v) re-measure compliance. For steps ii) and v), compliance with a standard was to be assessed for 25 women. Multilevel mixed effects logistic regression models were used to analyse data for all standards.

Standards-based audit was an effective method to improve the quality of care. The crude overall compliance rate rose from 45% in the control phase (pre-action in audit cycle) to 63% in the intervention phase (post-action). There was a statistically significant improvement in compliance for standards audited: the adjusted (for standard, facility type, month, and healthcare facility by month clustering) OR (95% CI) was 2.80 (1.65,4.76). The most frequently taken actions to improve compliance with standards included: i) providing support to improve staff performance to better carry out clinical duties and improve general conduct through re-orientation and staff meetings as well as improved supervision, and, ii) ensuring basic equipment and consumables were available on site (thermometers, rapid diagnostics, partograph).

Areas for improvement

Title:

The authors should write specific and concise title reflecting the study. I think no need to mention cluster randomized trial, they can add cross sectional study.

Abstract:

The authors should revise the language to improve readability.

The authors should select appropriate key words.

The authors should make sure that the abstract don’t exceed 300 words.

The authors should use IMRaD format to write the abstract.

Introduction:

The authors should revise introduction section for grammar issues and language to improve readability.

The authors used statistics in first paragraph of introduction section, they should mention from where this statistics came, which mean from which country.

The authors should join some related paragraphs together.

The authors should mention the gap clearly to the reader and the need for the study.

Overall

Material and Methods:

Study design, the authors should mention clearly the designed used and the important of the study design.

Trial intervention, clear.

Trial outcome, clear.

Study setting, The authors should revise the language to improve readability.

Developing of standards, clear.

Training on standards based audit, The authors should revise the language to improve readability.

Assigning standards, the authors should revise the language to improve readability and complete the missing information.

Data collection, the authors should mention if they used any tool to collect data. For the tool used in this study, they authors should mention from where they obtained this tool? Is it adopted or developed by the authors. If developed by authors how they check the validity and reliability of the tool?

Sample size, clear.

Data analysis, The authors should revise the langue to improve readability.

Ethical consideration, clear.

Results:

The authors should follow appropriate labelling of the title of each table in the manuscripts according to the journal’s guidelines. While the study appears to be sound, the language is unclear, making difficult to follow. Thus, the authors should work with a writing editor to improve readability of the text. The results section is clear.

References:

The authors should revise all references according to the guidelines provided.

Reviewer #3: The paper presented an interesting and important topic. However, the author needs to do some major revision.

- Can the description of Malawi level of health care delivery be included in the introduction? Emergency obstetric care take place at the secondary and tertiary level. The write up did not tell us at which level.

- The analysis also did not tell us what the adherence to this standard was at these two levels.

- The introduction also did not tell us whether both private and public facilities were used. Mission hospital was later mentioned (Line 290). This shows a mixed of the hospital types.

- How does the standard relate to the WHO quality of care (QoC) standards? Were any indicators adopted from the WHO QoC measurement?

- The table illustrating the 25 standards should have been added to the main document.

- The standard should have been categorized with relatable descriptive terms. This would have made the analysis easy to read and relate to.

- Under result, mentioning the standard that health facilities audited would have made the segment clear. (Line 269 -268). Line 362 -363 should be moved to the beginning of the analysis. It has basic information that would make the analysis clear.

- Line 309 mentioned all 12 standards. This contradict the 25 standard that was mentioned in line 29 -20.

- Line 415 – 434. The discussion did not adequately explain the analysis

- Line 427 -429 (The statement is not clear)

the author should review and rewrite the analysis and discussion.

6. PLOS authors have the option to publish the peer review history of their article (what does this mean?). If published, this will include your full peer review and any attached files.

Reviewer #1: No

Reviewer #2: **Yes: **Zalikha Khamis Al-Marzouqi

Reviewer #3: No

---

## [Author Response · Author response to Decision Letter 0]

13 Aug 2024

Reviewer 1:

Comment 1- Line 312: Provide a measure of dispersion for the mean of 20.6. Response: Thank you. This has been added.

Comment 2- Table 1: Define all abbreviations. Response: Thank you. This has now been done

Comment 3- Indicate if an effect for the repeated measures from individual patients was included in the logistic regression. These correlated measures need to be accounted for in the models. Response: Please note that there was no intention to measure any individual client multiple times, so no identification details were captured as indicated on lines 212-3. Therefore patient was not included in the model

Clusters (healthcare facilities) were however assessed repeatedly for each standard assessed and therefore models fitted did include effects for cluster and for interactions of cluster with Month (line 236) and with Standard (lines 386-9)

Comment 4- Line 378: The common abbreviation for standard deviation is SD. Response: Thank you this has been amended 

Comment 5- Cite the statistical software used for the analysis. Response: This has been added in line 229

Reviewer 2:

Comment Areas for improvement

Title:The authors should write specific and concise title reflecting the study. I think no need to mention cluster randomized trial, they can add cross sectional study. Response: Thank you for your suggestion. We have checked the requirements again: The title has 17 words (102 characters) which is within the requirements. 

CONSORT guidelines indicate that the design should be indicated in the title. Since the design is cluster randomized, we have retained this term in the title.

Comment Abstract: The authors should revise the language to improve readability. The authors should select appropriate key words.

The authors should make sure that the abstract don’t exceed 300 words.

The authors should use IMRaD format to write the abstract. Response: Thank you for your suggestions. We have revised the Abstract as best as possible within the word limit permitted (300 and the abstract is 296 words). 

IMRaD headings have been added and the format used by PLosOne has been followed. 

The authors should revise introduction section for grammar issues and language to improve readability. We have revised the Introduction and have again checked all English grammar and spelling throughout the manuscript. 

Comment The authors used statistics in first paragraph of introduction section, they should mention from where this statistics came, which mean from which country. Response: As stated in the opening sentence these statistics are global statistics as this seems most relevant in this manuscript and the Introduction.

Comment The authors should join some related paragraphs together. Response: Three paragraphs have been joined (now lines 93 -107)

Comment The authors should mention the gap clearly to the reader and the need for the study. Response: The gap (between availability and quality) is mentioned in the introduction (now 120-22). 

Comment Study design, the authors should mention clearly the designed used and the important of the study design. Response: The study design section (lines 129-138) identifies the trial design used and explains the reason this design was selected

Comment Study setting, The authors should revise the language to improve readability. Response: We have re-checked this section and revised the text to improve the language. 

Comment Training on standards based audit, The authors should revise the language to improve readability. Response: We have re-checked this section and revised the text to improve the language.

Comment Assigning standards, the authors should revise the language to improve readability and complete the missing information. Response: We have re-checked this section including language and clarified it further. 

Comment Data collection, the authors should mention if they used any tool to collect data. For the tool used in this study, they authors should mention from where they obtained this tool? Is it adopted or developed by the authors. If developed by authors how they check the validity and reliability of the tool? Response: The data collection section explains that standard-specific tools were developed for the study (lines 206-8). 

The wording has been revised to indicate more clearly that the tools used were developed specifically for each standard and for the study. 

Each tool captured data regarding the criteria to be fulfilled for there to be compliance with the relevant standard. 

Comment Data analysis, The authors should revise the langue to improve readability. Response: We have re-checked this section and clarified the language used.

Comment Results: The authors should follow appropriate labelling of the title of each table in the manuscripts according to the journal’s guidelines. While the study appears to be sound, the language is unclear, making difficult to follow. Thus, the authors should work with a writing editor to improve readability of the text. 

Response: Table titles have been revised to improve clarity of content and ensure they are brief and descriptive

We have carefully reviewed the text, made some changes to clarify details and hope we have now met your expectations of readability for a global audience.

Comment References: The authors should revise all references according to the guidelines provided. Response: We believe we have provided the references in line with the instructions for the journal, but did find a couple of errors which have now been corrected.

Reviewer 3:

Comment Can the description of Malawi level of healthcare delivery be included in the introduction? Emergency obstetric care take place at the secondary and tertiary level. The write up did not tell us at which level. Response: Thank you for these comments. We have included ‘levels’ in the Introduction section (hospital – secondary level and health centre – primary level) for Comprehensive and Basic EmONC respectively. 

In Malawi hospitals are generally considered secondary level and health centres are primary level. We have clarified this in the manuscript in the Methods section (setting) 

Comment - The analysis also did not tell us what the adherence to this standard was at these two levels. Response: Many thanks for this suggestion – we have now added statistics for both levels in the Results section and the Abstract

Comment - The introduction also did not tell us whether both private and public facilities were used. Mission hospital was later mentioned (Line 290). This shows a mixed of the hospital types. Response: There is no active private sector providing EmONC (either basic or comprehensive) in any of the 5 districts included in the study.

‘Mission’ hospitals provide public healthcare in the same was as hospitals that are run only by the Ministry of Health in Malawi.

We have amended the text to avoid any confusion.

Comment The main difference in the health care facility types was whether they were a hospital or a health centre and whether they were providing Basic or Comprehensive EmONC. We have provided these details in Supplementary Table 2. Thanks for this comment. 

Comment - How does the standard relate to the WHO quality of care (QoC) standards? Were any indicators adopted from the WHO QoC measurement? Response: Thank you for this question.

The WHO standards were used as reference but are not formatted to allow for audit. 

Comment - The table illustrating the 25 standards should have been added to the main document. Response: All 25 standards are provided in the Supplementary Material. (Supplementary Table 3) 

The 12 standards which were subjected to audit are listed in Table 1. 

The other 13 standards were not used so we do not think they need to be moved from the supplementary material into the manuscript as this would make the manuscript overly long.

Comment - The standard should have been categorized with relatable descriptive terms. This would have made the analysis easy to read and relate to. Response: Thank you for this feedback, standards are now referred to using terms that help identify them rather than numbers

Comment - Under result, mentioning the standard that health facilities audited would have made the segment clear. (Line 269 -268). Line 362 -363 should be moved to the beginning of the analysis. It has basic information that would make the analysis clear. Response: Information on the standards audited at each facility is provided in Supplementary Table 4.

Standards are now referred to by description rather than number.

As explained under Trial outcomes (lines 150-1) the specific standards audited were not of interest.

Comment - Line 309 mentioned all 12 standards. This contradict the 25 standard that was mentioned in line 29 -20. Response: There were 25 standards available from which to choose. Only 12 standards were actually audited; we have inserted ‘used’ on line 316 to clarify this

Comment - Line 415 – 434. The discussion did not adequately explain the analysis Response: We have amended the text as best as we can hopefully clarifying all points raised.

Many thanks for all the comments and critical reading of the manuscript.

Comment - Line 427 -429 (The statement is not clear) Response: Thank you for highlighting this - we agree that the statement could be clarified and have added further detail both in the discussion and the relevant results section.

Comment the author should review and rewrite the analysis and discussion. Response: We have amended the text as best as we can hopefully clarifying all points raised. Many thanks for all the comments and critical reading of the manuscript.

Editors comments:

Comment 1 Please ensure that your manuscript meets PLOS ONE's style requirements Thank you. The heading styles have been updated to comply with your requirements. 

Comment 2 Please state the name of the registry and the registration number . . . on the title page of your manuscript These details were included in submission data and are now included on the title page of the manuscript.

Comment 3 Please state what role the funders took in the study. The statement suggested is correct and is included in the cover letter.

Comment 4 Please include captions for your Supporting Information files at the end of your manuscript, and update any in-text citations to match accordingly. The supporting information has been restructured to comply with your requirements. The tracked changed manuscript contains the changes made to supplementary information but the clean version only contains the necessary captions. (Insertion of the supplementary material document was not tracked, but subsequent edits were).

Comment 5 Please review your reference list to ensure that it is complete and correct. These have been checked

---

## [Editor Report · Decision Letter 1]

9 Sep 2024

Standards-based audit to improve quality of maternal and newborn care – a stepped-wedge cluster randomised trial in Malawi

PONE-D-23-41107R1

Dear Dr. White

We’re pleased to inform you that your manuscript has been judged scientifically suitable for publication and will be formally accepted for publication once it meets all outstanding technical requirements.

Kind regards,

Mergan Naidoo, PhD

Academic Editor

PLOS ONE
---

## [Editor Report · Acceptance letter]

20 Sep 2024

PONE-D-23-41107R1 

PLOS ONE

Dear Dr. White, 

I'm pleased to inform you that your manuscript has been deemed suitable for publication in PLOS ONE. Congratulations! Your manuscript is now being handed over to our production team.

Kind regards, 

on behalf of

Professor Mergan Naidoo 

Academic Editor

PLOS ONE